# Tularemia Presenting Solely with Cervical Lymphadenopathy and Fever

**DOI:** 10.3390/diagnostics12082000

**Published:** 2022-08-18

**Authors:** Göran Ramin Boeckel, Jan Basri Adiprasito, Neele Judith Froböse, Frieder Schaumburg, Richard Vollenberg, Phil-Robin Tepasse

**Affiliations:** 1Department of Medicine D for Nephrology and Rheumatology, University Hospital Muenster, 48149 Muenster, Germany; 2Department of Medicine B for Gastroenterology, Hepatology, Endocrinology and Clinical Infectiology, University Hospital Muenster, 48149 Muenster, Germany; 3Institute of Medical Microbiology, University of Muenster, Domagkstraße 10, 48149 Muenster, Germany

**Keywords:** tularemia, lymphadenopathy, *Francisella tularensis*, tuberculosis, fever of unknown origin

## Abstract

A 52-year-old German female presented with cervical lymphadenopathy and fever. Despite the initial symptom-presentation leading to the consideration of sarcoidosis, lymphoma, tuberculosis, and toxoplasmosis, an extensive serologic and histo- and molecular pathologic workup eventually indicated a likely diagnosis of tularemia. This case brings to light that tularemia is a diagnostic challenge and requires high reliance on the epidemiological context thorough patient history, and an extensive interdisciplinary diagnostic workup.

## 1. Introduction

The Gram-negative coccobacillus *Francisella tularensis* causes the zoonosis tularemia [1,2]. Five subspecies exist of this pathogen, of which *F. tularensis* subsp. *tularensis* (called *F. tularensis* type A) and *F. tularensis* subsp. *holarctica* (called *F. tularensis* type B) cause infections in humans. The occurrence of *F. tularensis* covers much of the Northern Hemisphere; type A is common in North America, while type B is distributed in Europe and Asia [3]. The highest incidences of tularemia have been reported in Sweden, Finland, and Turkey [4]. In Germany, tularemia is a rare disease with an estimated incidence of 0.03 cases per 100,000 people per year [5].

Among others, the synonyms of tularemia include Francis disease, deer-fly fever, and rabbit fever. Tularemia presents with a wide range of clinical manifestations, from asymptomatic courses of disease to septic shock and death within weeks. The infection usually occurs after skin or mucosal contact with (the materials of) infected animals, or through invertebrate vectors (e.g., mosquitos and/or ticks), the inhalation of contaminated dust, or contaminated food (e.g., undercooked meat, must) and water [5,6,7]. Depending on the clinical expression, six manifestations of the disease (potentially overlapping) are distinguished: ulceroglandular, glandular, oculoglandular, pharyngeal, pneumonic, and typhoidal tularemia [8]. Within three weeks following pathogen exposure (a median of 3–5 days), patients present with non-specific, abrupt symptoms such as fever, fatigue, anorexia, headache, abdominal pain, emesis, or diarrhea [9].

The most common manifestation (ulceroglandular disease) usually presents with fever, lymphadenopathy, and a single erythematous skin lesion.

After direct pathogen-inhalation, pneumonic disease may present with a non-productive cough, peribronchial infiltrates, and/or pleural effusion.

Glandular tularemia displays without characteristic skin lesions and is found more often in children than in adults [10].

Especially due to the unspecific initial symptoms and the high reliance on the epidemiological context, tularemia is considered to be underdiagnosed [5,11]. Treatment mostly consists of antibiotic therapy. Streptomycin and gentamicin seem to be most effective; however, due to the better side-effect profile, quinolones (e.g., ciprofloxacin) and tetracyclines (e.g., doxycycline) appear to be an effective therapeutic alternative [12,13].

Here, we present the case of an adult woman with solely lymph node swelling and fever. Despite the initial concern about several differential diagnoses including tuberculosis, lymphoma, and toxoplasmosis, a thorough patient history, as well as an extensive interdisciplinary diagnostic workup, eventually revealed tularemia.

## 2. Case Presentation

A 52-year-old female was transferred to our tertiary care hospital for further evaluation of cervical lymph node swelling and suspicion of tuberculosis.

Two months prior to presentation at our hospital, the patient had been admitted to another hospital because of fatigue and fever. She had received antibiotic treatment with a cephalosporin (cefaclor) as well as ciprofloxacin from her general practitioner, without symptom alleviation. A diagnosis of sarcoidosis with lung affection (Table 1) had been considered; however, there were no conclusive radiological findings.

Because of the progressive cervical lymph node swelling (Figure 1A), the case had been presented to otolaryngology and a CT scan of the neck was performed. The findings showed centrally fused space-occupying processes on the left neck with additional left-sided enlarged and increased lymph nodes in the remaining cervical-to-supraclavicular stations, primarily classified as malignancy-suspicious (Table 1).

A panendoscopy with cervical lymph node extraction was performed. Histology found no signs of malignancy, but found scattered multinucleated giant cells. Despite negative Ziehl–Neelsen staining, tuberculosis (TBC) was still considered a likely diagnosis based on the pathology. Therefore, further molecular diagnostics were initiated (Table 1), which did not reveal any conclusive findings.

An additional radiological analysis (CT scan chest/abdomen) showed no signs of primary tumors or metastases. However, isolated granulomas in the left lower lung were described. Postoperatively, the cervical wounds showed prolonged secretion and painful surrounding edema (Figure 1B). The patient was ultimately released with oral antibiotics (amoxicillin/clavulanic acid), and a recommendation for further evaluation at our infectious diseases department was made.

Upon presentation at our hospital, the only remaining symptom was cervical lymph node swelling. The last time B-symptoms (fever, night sweats, weight loss) had occurred had been two months before presentation. The patient presented with normal vegetative status without any signs of infection. An extended patient history revealed that she was a never-smoker and rarely consumed alcohol. She was born in Poland and moved to Germany 40 years ago. She was the mother of three children and lived with her mother-in-law and her husband. Her occupation was as a cook and her last vacation abroad had been to Poland two years prior to her symptom onset. The patient had cats as pets.

Physical examination showed equal pupils with a normal reaction to light and no alopecia. The dental status was regular with rosy, moist oral mucosa, no plaque, no lesions, and no petechiae. The heart and lungs were normal upon auscultation. The abdominal wall was soft, the liver and spleen were not enlarged and palpable, with no resistance. The left cervical lymph nodes showed persistent lymphadenopathy with scaring tissue (after biopsy); there was no palpable peripheral lymphadenopathy of the clavicular, axillary, or inguinal region. The joints were freely mobile, without edema. The skin showed signs of erythema/exanthema without signs of bleeding. There was no evidence of a focal neurological deficit.

The standard laboratory analyses were unremarkable with minor normochromic anemia (10.8 g/dL (normal range: 11.9–14.6 g/dL)) and minor LDH elevation (290 U/L (normal range: 135–214 U/L)). Screening tests for HIV and Hepatitis B and C were negative.

An evaluation of external radiological scans in our department provided no additional insights.

An interferon-gamma release assay (Table 1) was used to test for exposure to *Mycobacterium tuberculosis* complex and showed a negative result. Because of the pets, a differential diagnosis of toxoplasmosis (Table 1) or bartonellosis (Table 1) was considered. The findings were consistent with latent toxoplasmosis (toxoplasma-IgG 147 IU/mL (normal range: <10 IU/mL); IgM negative) and a negative bartonellosis serology was found.

After new swelling at the operation site due to abscesses, needle aspiration was performed and new bioptic material was obtained. For bacterial cultures, Columbia blood agar, chocolate agar and Schaedler KV agar were incubated (Table 1). As the culture remained without pathogen detection, a broad molecular diagnostic was initiated. This included diagnostics for tuberculosis, atypical mycobacteria, *Bartonella henselae*, *Cryptococcus neoformans*/*Cryptococcus gattii*, *Aspergillus* sp. And other fungi, *Chlamydia trachomatis*, and *Brucella* sp., as well as a PCR diagnostic, which detects pathogens of the (eu-) bacteria domain by means of standard in-house PCR with amplicon detection in agarose gel [14] (Table 1). *F. tularensis* subsp. *holarctica* was thereby identified via 16S RNA gene sequence comparison (GenBank accession no. CP044005.1, identity 99.72%; Table 1). All the other tests showed negative results. The DNA and associated native material were subsequently sent to the tularemia consulting laboratory in Berlin, Germany, where the molecular result was confirmed, but cultivation was not successful. Serological testing for tularemia was sent to the Bernhard Nocht Institute for Tropical Medicine (Hamburg) and showed positive results (Ig ELISA; IgG Blot; no further quantification).

In another in-depth patient history, the patient reported buying venison from a hunter on a regular basis. Most recently, in the month prior to the onset of symptoms, she had consumed venison that she had purchased from the local hunter approximately one year earlier and had stored in a frozen state until consumption. As she was the only one to touch the uncooked meat, none of the relatives who ate the cooked meat became sick. Unfortunately, no frozen samples of the suspected meat were left over for examination. Therapy with ciprofloxacin twice daily for 3 weeks was initiated.

Upon follow-up at our outpatient clinic after completion of the treatment, the left cervical operation site had improved significantly, with distinct sonographic regression of the lymphadenopathy (Figure 1C). Five months after the initial symptom onset, the patient reported being well and free of symptoms.

## 3. Discussion

The previously described nonspecific clinical presentation of tularemia simplifies the misdiagnosis due to the multitude of differential diagnoses. Regarding the nonspecific lymphadenopathy, numerous differential diagnoses, such as, among others, streptococcal or staphylococcal infections, brucellosis, bartonellosis, sporotrichosis, *Pasteurella multocida* infections, and parotitis in mumps, should be considered [15].

Furthermore, symptoms consisting of fever and a general feeling of illness in combination with regional lymph node swelling may indicate lymphoma [16].

Granuloma-like changes in the lungs, in particular, can point toward tuberculosis [17]. Other pulmonary diseases may also present with a similar clinical picture. These include abscessing pneumonia, but also mycoplasma pneumonia, pneumonic plague, or mycoses [15]. The most important differential diagnoses and the possibilities of differentiation are listed in Table 1.

In general, the blood count and laboratory chemistry, if any, show nonspecific changes such as moderate leukocytosis or CRP elevation at the onset of the disease. Furthermore, mildly elevated levels of lactate dehydrogenase (LDH), alkaline phosphatase (AP), and serum transaminases are possible. Furthermore, patients may present with myoglobinuria as well as pyuria, which may be falsely diagnosed as a urinary tract infection. Rhabdomyolysis with elevated serum creatinine kinase levels has also been described in some patients [15,18,19].

Patients with lymphadenopathy and nonspecific symptoms should also be examined for tularemia if they have a history of potential exposure. Furthermore, patients with suspected tuberculous lung disease or a granulomatous disease should also be considered for tularemia, especially if infection with mycobacteria is excluded [17].

The diagnosis itself is mainly made via molecular methods. Obtaining a culture requires containment level 3, and is therefore (in Germany) reserved for specialized laboratories; the fastidious pathogen grows on cysteine-containing solid culture media (blood or chocolate agar) at 37 °C under aerobic conditions [20,21].

Another possibility for direct pathogen detection is the detection of DNA via PCR analysis. Data from 2008 showed increased sensitivity in PCR diagnostics compared to cultural detection of the pathogen [22].

Serology is the third pillar of the diagnosis of *Francisella tularensis* infections. A titer of more than 1:160 or a 4-fold increase in the titer in successive samples has diagnostic value. The advantage of serological testing, however, is the comparatively simple procedure and the low number of false-negative results [23]. However, cross-reactions, especially with Brucella and Yersinia species, must be mentioned, as these can lead to false-positive results [24].

In the context of cervical lymphadenopathy, cytology after fine-needle aspiration is also considered to have a potential diagnostic role in addition to serology, PCR, and culture [25]. Furthermore, flow cytometric assays looking at the relative number of CD3+/CD4−/CD8− T cells could play a role in the early diagnosis of tularemia. These assays showed elevated levels 7 days before a serological diagnosis was possible [26].

## 4. Conclusions

The diagnosis of tularemia is an iterative process due to the rarity of the disease and the non-specific symptoms. Regarding patient history, it is important to pay attention to the patient’s origin from areas with a high incidence of tularemia, as well as to the fact that the patient belongs to a population group that has contact with wild animals or their processing, and therefore, has an increased probability of exposure.

This case report demonstrates the relevance of broad molecular diagnostics in unclear infectious cases. Evidence of the pathogen was provided via 16S RNA gene sequencing in a case where cultural cultivation was no longer possible.

A broad interdisciplinary diagnostic approach by clinicians and in microbiology, pathology, and radiology is essential.

## Figures and Tables

**Figure 1 diagnostics-12-02000-f001:**
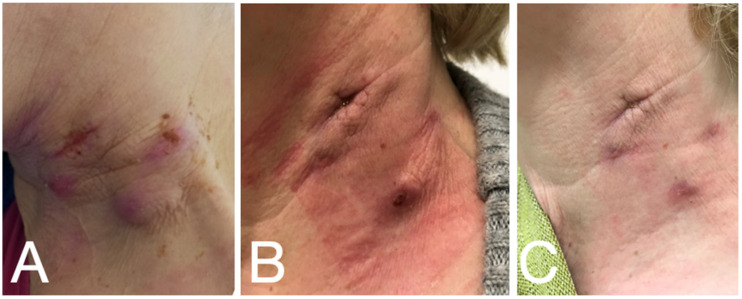
**Clinical course of the lymph node extirpation.** (**A**) initial presentation; (**B**) upon transfer to our hospital; (**C**) one month after treatment with ciprofloxacin.

**Table 1 diagnostics-12-02000-t001:** Considered differential diagnoses, and their diagnostic approaches.

Differential Diagnosis	Diagnostic Approach	Specimen	Details
Sarcoidosis	Chest X-ray		Admitting clinic
Carcinoma metastasis, lymphoma	CT scan of the neck Panendoscopic biopsy		Admitting clinic
Tuberculosis	Panendoscopic biopsy		
	PCR Ziehl–Neelsen stain TBC culture	Abscess fluid; sputum	Xpert MTB/RIF Ultra, Cepheid, Krefeld, Germany; solid and liquid culture (Löwenstein–Jensen agar (BD, Heidelberg, Germany), Stonebrink agar (Oxoid, Wesel, Germany), and MGIT (BD))
	Interferon-gamma release assay	Lithium heparin blood	QuantiFERON IFN-γ Standard, Qiagen, Hilden, Germany
Atypical mycobacteriosis	PCR TB culture	Abscess fluid	“In-house“-PCR *Mycobacterium* genus; solid and liquid culture (Löwenstein–Jensen agar (BD), additionally incubated at 30 °C, Stonebrink agar (Oxoid), and MGIT (BD))
Toxoplasmosis	ELISA PCR	Blood serum; abscess fluid	SERION ELISA classic, *Toxoplasma gondii* IgG/IgM, Virion Serion, Würzburg, Germany; ProGenie RealCycler TGON-U/TGON-G, Valencia, Spain
Bartonellosis	IIFT PCR	Blood serum; abscess fluid	Anti-Bartonella-henselae-IIFT (IgG and IgM), Euroimmun, Lübeck, Germany; ProGenie RealCycler BART-U/BART-G
Cryptococcosis	Cryptococcal antigen lateral flow assay	Abscess fluid	CrAg LFA, IMMY Inc., Norman, USA
Chlamydiosis	PCR	Abscess fluid	Chlamydia trachomatis PCR Kit, GeneProof a.s., Dolní Heršpice, Czech Republic
Brucellosis	PCR	Abscess fluid	BactoReal Kit Brucella spp., Ingenetix, Vienna, Austria
	Bacterial cultures	Abscess fluid	Columbia blood agar and Chocolate agar (incubated at 5% CO_2_ at 36 ± 1 °C for 48 h under aerobic conditions); thioglycolate broth (all BD); Schaedler KV selective agar (incubated under anaerobic conditions at 36 ± 1 °C; Oxoid)
	(Eu-) bacteria PCR	Abscess fluid	(Eu-) bacteria domain, standard in-house PCR

## Data Availability

Not applicable.

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
