# Peer review of "Tularemia Presenting Solely with Cervical Lymphadenopathy and Fever"

_diagnostics, 2022, doi:10.3390/diagnostics12082000_

Round 1

Reviewer 1 Report

Brief Summary

This is a case report of a female patient who was transferred to a tertiary care hospital for further evaluation. At the time she was admitted her only symptoms were lymph node swelling and fever.  Two months prior to this, she had been admitted to a hospital because of fatigue and fever with a potential diagnosis for sarcoidosis She had been on antibiotic treatment given to her by her general practitioner without symptom alleviation.

At the previous hospital she had undergone many tests and procedures. Following a biopsy of the swollen lymph node, there was no sign of a malignancy and the only histological finding was scattered multinucleated giant cells. There was evidence of isolated granulomas in the lower left lung, and despite a negative TB test, tuberculosis was a presumptive diagnosis.

Upon arrival at the tertiary hospital, her only remaining symptom was cervical lymph node swelling. Multiple tests and scans provided no additional insights. Cultures of material obtained from an additional biopsy of abscesses that had formed at the previous operation site were also negative. Finally a broad molecular diagnostic was initiated and F. tularensis subsp. holarctica was identified by 16S gene sequence comparison. In depth patient history revealed that the patient had bought venison from a hunter on a regular basis. She was immediately placed on ciprofloxacin for 3 weeks.

Review:

Authors in the introduction gave a comprehensive general overview of tularemia. They have pointed out the importance of taking a thorough in-depth history and the need to pay attention to details regarding origin and potential for contact with wild animals or their carcasses. And due to the non-specific symptoms, tularemia should be considered once tuberculosis is ruled out.

This case report also points out the importance of molecular diagnostics in unclear infectious cases and demonstrated that a braod multidisciplinary diagnostic approach is essentails.

Author Response

We thank the reviewer for the positive review of our manuscript and we are very happy that our publication intent/message was conveyed.

Reviewer 2 Report

This case report is well written, easy to follow and a good and an excellent reminder of keeping in mind also the rare diseases. Zoonotic infections are increasing in incidence and currently we have better tools to detect them. I only have one question: the patient received already initially ciprofloxacin, but why did it not resolve the infection? Did this affect the following diagnostic workup and did it cause delays in reaching the correct diagnosis?

In addition, you mention the vector-borne route of infections, I would mention mosquitoes as a relevant vector, at least in Northern Europe that is far more relevant than ticks.

Author Response

Thank you for the review.

Please, find our detailed response attached.

Kind regards

Göran Boeckel
